# LARGE TRAJECTORY MODELS ARE SCALABLE MOTION PREDICTORS AND PLANNERS

## ABSTRACT

Motion prediction and planning are vital tasks in autonomous driving, and recent efforts have shifted to machine learning-based approaches. The challenges include understanding diverse road topologies, reasoning traffic dynamics over a long time horizon, interpreting heterogeneous behaviors, and generating policies in a large continuous state space. Inspired by the success of large language models in addressing similar complexities through model scaling, we introduce a scalable trajectory model called State Transformer (STR). STR reformulates the motion prediction and motion planning problems by arranging observations, states, and actions into one unified sequence modeling task. With a simple model design, STR consistently outperforms baseline approaches in both problems. Remarkably, experimental results reveal that large trajectory models (LTMs), such as STR, adhere to the scaling laws by presenting outstanding adaptability and learning efficiency. Qualitative results further demonstrate that LTMs are capable of making plausible predictions in scenarios that diverge significantly from the training data distribution. LTMs also learn to make complex reasonings for long-term planning, without explicit loss designs or costly high-level annotations.

## 1 INTRODUCTION

Motion planning and prediction in autonomous driving rely on the ability to semantically understand complex driving environments and interactions between various road users. Learning-based methods are pivotal to overcoming this complexity as rule-based and scenario-specific strategies often prove inadequate to cover all possible situations and unexpected events that may occur during operations. Such learning problems can be regarded as conditional sequence-to-sequence tasks, where models leverage past trajectories to generate future ones, depending on the observations. Notably, these problems share structural similarities with other sequence modeling problems, such as language generation. This parallel in terms of both the problem settings (i.e. predicting the next state in a sequence vs. predicting the next token of language in a sequence) and the challenges (i.e. learning diverse driving behaviors against observations vs. answers against prompts) evokes a compelling question: Can we follow large language models to crack these complexities of motion prediction and motion planning in autonomous driving by large trajectory models?

Recent studies (Mirchandani et al., 2023; Zeng et al., 2023) have demonstrated that the LLMs excel not only in natural language generation but also in tackling a wide range of sequence modeling problems and time series forecasting challenges. Building on these insights, prior research (Chen et al., 2021; Janner et al., 2021; Sun et al., 2023) have effectively utilized conditional causal transformers to address motion planning as a large sequence modeling problem, with both behavior cloning and reinforcement learning. Furthermore, (Brohan et al., 2023) replace the transformer backbone with language models, demonstrating the potential to merge motion planning along with other modalities within one large sequence for LLMs.

However, learning from large-scale real-world datasets for autonomous driving presents additional complexities: (i) intricate and varied map topology, (ii) prediction within a substantially large continuous space, (iii) noisy and ambiguous demonstrations from diverse human drivers, and more. Furthermore, the evaluation metrics associated measure accuracy up to 8 seconds, placing additional challenges on the long-term reasoning capabilities of different methods. Previous works, like (Seff et al., 2023), fail to scale the size of the model over 10M before overfitting.

In this study, we propose a novel and scalable adaptation of conditional causal transformers named State Transformers (STR). Our experimental results indicate that scaling the GPT-2 model backbone (Radford et al., 2019) significantly improves the learning efficiency and mitigates generalization problems in complex map topologies when arranging all components into one sequence for learning. In this sequence, we arrange the embeddings of encoded maps, past trajectories of road users, and traffic light states as conditions for future state sequence generation. The design of STR is both concise and adaptable. STR is compatible with additional supervisions or priors by inserting additional embeddings, like adding prompts for the language generation models. Specifically, we introduce Proposals followed by Key Points in the sequence to alleviate the problem of large output spaces. Proposal classification has been proven (Shi et al., 2022) as a useful technique to guide the model to learn from a mixed distribution of different modalities on single agents, easing the problem of ambiguous demonstrations and large output spaces. Key Points serve a dual purpose: on one hand, they improve the model's capacity for long-term reasoning during training, and on the other hand, they function as anchors to empower the model to generate high-level instructions and rules, such as lane changes, by incorporating them as additional generation conditions at different time steps. Lastly, we implement a diffusion-based Key Point decoder to fit the multimodal distribution of futures caused by the interactions of multiple road users.

We evaluate STR through rigorous experiments on two expansive real-world datasets: NuPlan (H. Caesar, 2021) for the motion planning task, and Waymo Open Motion Dataset (WOMD) (Ettinger et al., 2021) for the motion prediction task. The NuPlan dataset covers over 900 hours of driving experiences across four distinct cities located in two countries. Notably, the training subset of this dataset includes over 1 billion human-driving waypoints of the planning vehicle, aligning with the scale of vast language datasets. To comprehensively evaluate the scalability of our approach, we conducted experiments spanning 3 orders of magnitude in terms of the size of the training set and 4 orders of magnitude with respect to the model size. The model size is measured in terms of the number of trainable parameters within the GPT-2 backbone. The empirical results reveal intriguing parallels in the scaling behaviors with training the LLMs and the LTMs. Furthermore, we observed that LTMs exhibit superior performance when tested with previously unseen map scenarios, surpassing their smaller counterparts. These parallels between future state prediction challenges and the rapidly evolving field of language modeling tasks suggest promising opportunities for leveraging emerging language model architectures to advance the learning efficiency of motion prediction and motion planning in the future. Source codes and models are available at https://github.com/Tsinghua-MARS-Lab/StateTransformer.

## 2 RELATED WORKS

Motion planning constitutes a substantial research domain for autonomous driving. In this study, our primary focus lies on the learning part, recognizing that learning constitutes merely a segment of the comprehensive puzzle. More precisely, we approach the motion planning problem by framing it as a future state prediction problem. In the subsequent parts of this paper, we will use the terms "trajectory generation", "trajectory prediction", "state prediction", and "pose prediction" interchangeably. All these terms refer to the same objective, which is to predict the future states, including both the position and the yaw angle, of the ego vehicle.

**Motion Prediction.** In recent years, significant advancements have been made in learning-based trajectory prediction for diverse types of road users. Graph neural networks have been introduced as an effective way to encode the vectorized geometry and topology of maps (Gao et al., 2020; Liang et al., 2020). Given the multimodal nature of future motions, various decoder heads have been proposed, including anchor-based (Chai et al., 2019), goal-based (Zhao et al., 2021), heatmap-based models (Gu et al., 2021; Gilles et al., 2021). Transformer architectures have also been applied in motion prediction tasks to enable simple and general architectures for early fusion (Nayakanti et al., 2023) and joint prediction and planning (Ngiam et al., 2021). To provide a more comprehensive understanding of the interactions between multiple agents, there have been advancements including factored (Sun et al., 2022) and joint future prediction models (Luo et al., 2023).

**Motion Planning.** Imitation learning (IL) and reinforcement learning (RL) are two major learning paradigms in robotics motion planning. IL has been applied to autonomous driving since the early

days (Pomerleau, 1988; Bojarski et al., 2016; Zhang et al., 2021; Vitelli et al., 2022; Lioutas et al., 2022). However, IL suffers from the covariate shift problem (Ross et al., 2011), where the accumulated error leads to out-of-distribution scenarios, posing severe safety risks. Reinforcement learning avoids the covariate shift problem by learning in a closed-loop simulator. Nevertheless, reinforcement learning introduces several unique challenges: reward design, high-quality traffic simulation, and sim-to-real transfer. It is especially challenging to design proper reward functions reflecting intricate human driving courtesies. More recently, (Hester et al., 2018; Vecerik et al., 2017; Lu et al., 2022) proposed to combine imitation learning and reinforcement learning to improve the safety and reliability of the learned policies.

**Diffusion Models in Prediction and Planning.** Diffusion models (Ho et al., 2020; Dhariwal & Nichol, 2021; Rombach et al., 2022) have emerged as a new paradigm for generative tasks. Recently, diffusion models have been used in motion planning tasks and have shown a strong ability to fit multimodal distributions with uncertainty. In the area of robotics, (Janner et al., 2022) utilizes diffusion models for motion planning by iteratively denoising trajectories and (Chi et al., 2023) proposes a method to learn policies in a conditional denoising diffusion process. As for trajectory prediction, (Jiang et al., 2023) utilizes diffusion models to capture the multimodal distribution for multi-agent motion prediction in autonomous driving and achieves state-of-the-art performance.

**Scaling Laws in Large Language Models.** Hestness et al. (Hestness et al., 2017) identified scaling laws for deep neural networks. Kaplan et al. (Kaplan et al., 2020) conducted an extensive study of scaling laws in large language models. They empirically demonstrated that the performance of transformer-based models on a range of natural language processing tasks scales as a power-law function with the number of model parameters, amount of training data, and compute resources used for training. This research laid the groundwork for understanding how scaling influences the efficacy of language models. Henighan et al. (Henighan et al., 2020) further studied scaling laws in autoregressive generative modeling of images, videos, vision-language, and mathematical problems. These works suggest that scaling laws have important implications for neural network performance.

## 3 PRELIMINARIES AND PROBLEM SETUP

### 3.1 TRAJECTORY GENERATION

Motion prediction and planning are two classic problems in autonomous driving, often formulated in different setups. The goal of motion prediction is to estimate the possible future trajectories of other road agents. More specifically, given a static scene context $c$ such as a high-definition map, ego vehicle states $s_{\text{ego}}$ and a set of observed states of other road agents $\mathbf{s} = [\mathbf{s}_0, ..., \mathbf{s}_{N-1}]$, our goal is to predict the future states of other agents up to some fixed time step T. The overall probabilistic distribution we aim to estimate is $P(\mathbf{s}^T | c, s_{\text{ego}}, \mathbf{s})$.

On the other hand, the definition of motion planning is as follows: given scene context $c$, ego states $s_{\text{ego}}$, the observed states of other road agents $\mathbf{s}$, and a route $r$ from a navigation system, find the policy $\pi$ for our ego agent that maximizes expected return $R$, which is usually defined as a mixture of safety, efficiency, and comfortness: $\arg\max_{\pi} R(\pi | c, s_{\text{ego}}, \mathbf{s}, r)$. If we are given expert demonstrations $\pi^*$ in motion planning, and take the behavior cloning approach, then its formulation becomes estimating the probabilistic distribution of expert policy $P(\pi^* | c, s_{\text{ego}}, \mathbf{s}, r)$. This objective is highly similar to motion prediction, apart from taking an additional route as input. Therefore, it is reasonable to leverage a general framework to model both of them as trajectory generation tasks.

### 3.2 CONDITIONAL DIFFUSION MODELS

The primary goal of conditional diffusion models on trajectory generation tasks is to generate the future states conditioning on given context. Specifically, given a condition vector $\mathbf{f}$ and a desired output vector $\mathbf{s}^*$, the probabilistic framework that captures this relationship is defined as $P(\mathbf{s}^* | \mathbf{f})$, which is the probability distribution the diffusion model estimates. In this work, we let $\mathbf{s}^*$ be a latent feature related to $\mathbf{s}^{\mathbf{T}}$ (for example, Key Points of the future trajectory) and $\mathbf{f}$ be the feature obtained by forwarding $c, s_{\text{ego}}, s, r$ through a backbone (for example, certain Transformer backbone).

Particular conditional Denoising Diffusion Probabilistic Models (Ho et al., 2020) are based on two processes: the forward process (diffusion) and the reverse process (denoising). In the forward pro-

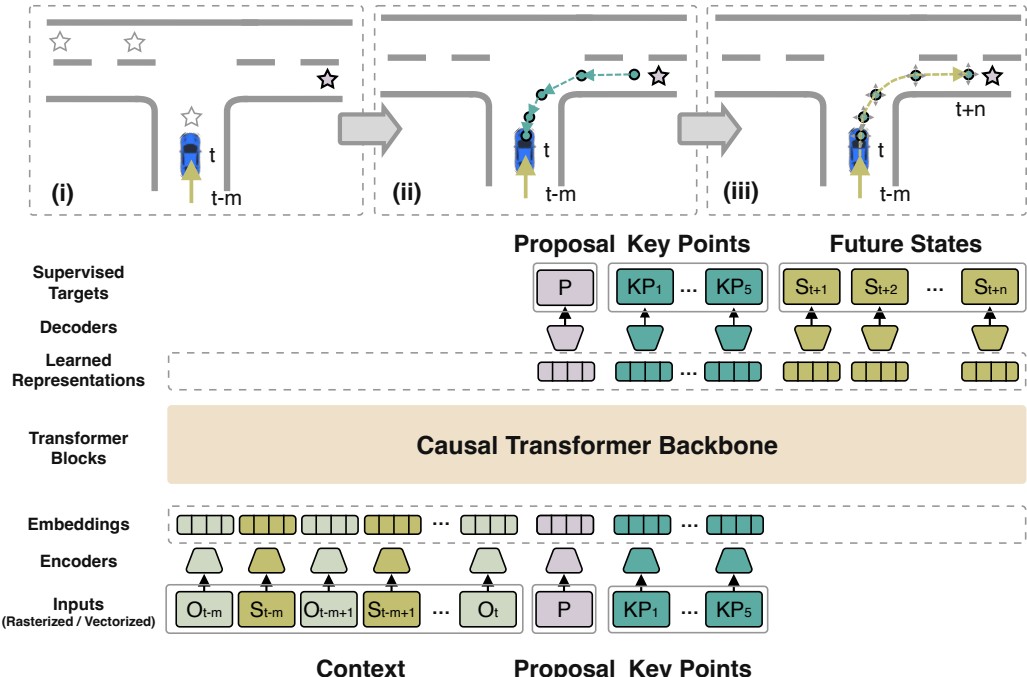

Figure 1: The architecture of STR. There are four components in the sequence, namely Context, Proposal, Key Points, and Future States. Each part is encoded by its corresponding encoder. The causal transformer backbone, the GPT-2 model in our experiments, learns representations on the embedding level. The Proposal and Key Points are two optional components. A full generation process of STR is as follows. (i) STR selects Top K Proposals indicating future directions. (ii) STR generates a set of Key Points consecutively. (iii) STR generates the future states.

cess $s^*$ is gradually transformed into noise tensor $\mathbf{z}$ through $T_d$ diffusion steps. At each diffusion time step $t_d$, the data point is perturbed with Gaussian noise $\varepsilon_{t_d}$:

$$s_{t_d}^* = \sqrt{1 - \beta_{t_d}} \cdot s_{t_d-1}^* + \sqrt{\beta_{t_d}} \cdot \varepsilon_{t_d} \tag{1}$$

where $s_0^* = s^*$, $s_{T_d}^* = z$ and $\beta_{t_d}$ is a noise schedule parameter. It can be derived to:

$$s_{t_d-1}^* = \frac{1}{\sqrt{1 - \beta_{t_d}}} \cdot \left( s_{t_d}^* - \sqrt{\beta_{t_d}} \cdot \varepsilon_{t_d} \right). \tag{2}$$

In the denoising phase, the goal is to generate $s^*$ when sampling from $z$. The key problem here is to estimate the noise term $\varepsilon_{t_d}$ with respect to $\mathbf{s}_{t_d}, \mathbf{f}, t_d$. Given an estimated term $\hat{\varepsilon}_{t_d}$, $s_{t_d-1}^*$ can be calculated with Eqn. (2). By performing this reverse process iteratively we can generate $s_0^*$ starting with $s_{T_d}^* = z$. Notably, by learning to estimate $\varepsilon_{t_d}$ through $\hat{\varepsilon}_{t_d} = g_\theta\left(\mathbf{s}_{t_d}^*, \mathbf{f}, t_d\right)$, the diffusion model captures the multi-modality structure inherent in the data distribution.

## 4 STATE TRANSFORMERS

In this section, we introduce each element in the sequence of our proposed approach STR for both motion planning and motion prediction tasks, shown in Fig. 1. The main idea is to formulate the future states prediction problem as a **conditional sequence modeling** problem. This formulation arranges the observations, the past states, and the future states into one large sequence. This formulation is maintained as general, allowing it to be applicable for both motion planning and motion prediction tasks with diverse intermediate supervisions or priors.

**Context.** To begin with, we define the Context, including the map, the past trajectory of other road users, and states of traffic lights, as observations at that time step and the past states of the ego vehicle. In the sequence generation problem setting, we treat these factors as conditions for

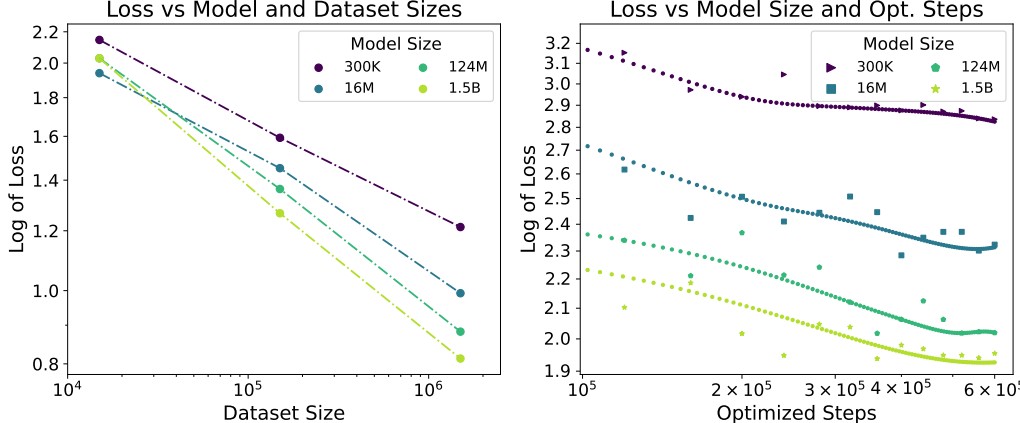

Figure 2: These two figures demonstrate the substantial scalability of STR, illustrating the scaling laws for training LTMs. The left figure reveals that LTMs exhibit smooth scalability with the size of the training dataset. When the training is not constrained by the size of the dataset, larger trajectory models tend to converge to a lower evaluation loss. The right figure shows that larger trajectory models learn faster to converge than their smaller counterparts, indicating superior data efficiency.

the generation, like prompts for language generation. The context can be both rasterized (tested later with NuPlan) or vectorized (tested later with WOMD) features, each paired with its respective encoder.

**Proposal.** Given the multi-modal distribution mixture of future states, we introduce a classification of the Proposals to separate along different modalities given the same prompt. The modalities here indicate different intentions in the future, such as changing speeds or changing lanes to the left or right. In this work, we leverage the intention points (Shi et al., 2022) as the Proposals. Ground truth Proposals are fed into the sequence facilitating a teacher-forcing style of training to guide the prediction of future states.

**Key Points.** Moreover, we introduce a series of Key Points into the sequence. These Key Points represent specific points at designated time steps from the future trajectory. They are trained in an auto-regressive manner with $P(K^M) = \prod_{m=1}^{M} P(K_m|K_{1:m})$ where $m$ is the number of the Key Points to predict. STR is compatible with diffusion decoders, if necessary, to cover the multi-modality at each time step. These Key Points guide the model to reason about long-term decisions during training.

During the generation process, post-procession, filtering, and searching can be applied to refine or select the optimal Key Points. In other words, these Key Points function as anchors for generating future states following additional high-level guidance. For example, if Key Points are off-road during the generation process, we can move them to the nearest lane to instruct future states to stay close to the road. For our experiments in this work, we select 5 Key Points in an exponentially decreasing order which are the states of 8s, 4s, 2s, 1s, and 0.5s in the future.

**Future States.** Finally, the model learns the pattern to generate all Future States given the Contexts, Proposals, and the Key Points with direct regression losses in a specified frequency, which is 10Hz for both WOMD and NuPlan in our experiments.

**Causal Transformer Backbone.** Our architecture is compatible with most transformer-based language models, serving as the causal transformer backbone. The backbone can be easily updated with the later iterations, or be scaled up or down by modifying the number of the layers. As a result, we can evaluate different sizes of backbones without touching the structure of the encoders or decoders for a fair comparison. Considering the excellent generalization ability and scalability, we test multiple language models and select the GPT-2 model as our Causal Transformer Backbone to report in this work.

## 5 EXPERIMENTS

We design experiments to (i) verify the scalability of STR, (ii) compare the performance with previous benchmarks, and (iii) implement ablation studies on design choices. To test the scalability of our approach, we train and evaluate STR on the motion planning task with NuPlan v1.1. The training dataset scales from 15k to 1 million scenarios and the models scale from 300k to 1.5 billion trainable parameters. Following (Kaplan et al., 2020), we measure the performance with converged evaluation losses, which are the lowest evaluation losses after training with sufficient computations. We train each size of our model with the full training dataset and compare the evaluation results with the performance reported in (Dauner et al., 2023). For the task of motion prediction, the model is trained with the size of 16 million with the training dataset of the WOMD v1.1 and compares the evaluation results with the performance from (Ngiam et al., 2021) and (Shi et al., 2022).

### 5.1 DATASETS AND METRICS

**Dataset for Planning.** We evaluate the task of motion planning on NuPlan Dataset and test the scalability of our approach on this dataset considering it is the largest dataset at the time. Following the setting of (Dauner et al., 2023), which is the current benchmark method on NuPlan's leaderboard, we use the same scenarios in the training set. Each frame in these scenarios is labeled with scenario tags, such as "turning left", "turning right", etc. This training set has 15,523,077 samples of 8 seconds driving across 4 different cities. For testing, we follow the previous setting to test on the validation 14 set for a fair comparison. The validation 14 set is a hand-picked subset of the whole validation set, which is claimed by (Dauner et al., 2023) to be well-aligned with the full validation set and the test set.

**Metrics for Planning.** For the motion planning task, NuPlan provides three sets of metrics including Open-Loop Simulation (OLS), Closed-Loop Simulation with Reactive Agents (CLS-R), Closed-Loop simulation with Non-Reactive Agents (CLS-NR). As discussed in (Dauner et al., 2023), there is a notable misalignment between the closed-loop simulation metrics and ego-forecasting. Since we focus on the learning performance of the models in this paper, we report and compare scores of the metrics on the accuracy, including averaged displacement error (ADE), final displacement error (FDE), miss rate (MR), and the OLS score. The miss rate in NuPlan metrics is the percentage of scenarios with a length of 15 seconds in which the max displacement errors in the three horizons are larger than the corresponding threshold. The OLS is defined as an averaged overall score of all the errors on displacement, heading errors, and the miss rate. Please refer to Appendix A.3 for more details.

**Dataset and Metrics for Prediction.** For the motion prediction task, we conduct training and testing on WOMD. For comparison, following previous approaches, we report testing results on the validation set with 4 metrics: mAP, minADE, minFDE, and MR. The MR here is the percent of scenarios in which the prediction's FDE is larger than a certain threshold.

### 5.2 ENCODERS AND DECODERS

**Encoders.** For the NuPlan dataset, we employ a raster encoder utilizing ResNet18 (He et al., 2016) because it is convenient for fast visualization and debugging. For the WOMD dataset, a vector encoder identical to MTR (Shi et al., 2022) is implemented. Additionally, we employ maxpooling along the instance axis instead of the time axis in order to arrange all the states in a sequence of embeddings to the backbone. For the Key Points, we implement a single Multilayer Perceptron (MLP) layer followed by an activation layer of Tanh. The ground truth Key Points are fed into the backbone during training to learn in an auto-regression style, implying predicting the next key point only. At the inference stage, the model rolls out the Key Points one by one. We use the same encoder for the Proposals on WOMD.

**Decoders.** STR involves three distinct decoders for the Proposal, Key Points, and Future States. We implement STR(CPS) which includes the Proposal for the motion prediction task on WOMD and STR(CKS) which includes the Key Points for the motion planning task on NuPlan. The Proposal decoders are two separate heads of MLPs respectively trained with a Cross-Entropy loss and an MSE loss. The Key Points decoders are diffusion decoders trained with an MSE loss. The Future State decoders are also MLPs trained with an MSE loss. For better convergence, we apply the diffusion process to the latent feature space of the Key Points rather than directly to the latent feature space of

Table 1: Performance comparison of motion planning on the validation 14 set of the NuPlan dataset.

| Methods | 8sADE ↓ | 3sFDE ↓ | 5sFDE ↓ | 8sFDE ↓ | MR ↓ | OLS ↑ |
|---|---|---|---|---|---|---|
| IDM (Treiber et al., 2000) | 9.600 | 6.256 | 10.076 | 16.993 | 0.552 | 37.7 |
| PlanCNN (Renz et al., 2022) | 2.468 | 0.955 | 2.486 | 5.936 | 0.064 | 64.0 |
| Urban Driver (Scheel et al., 2022) | 2.667 | 1.497 | 2.815 | 5.453 | 0.064 | 76.0 |
| PDM-Open (Dauner et al., 2023) | - | - | - | - | - | 72.0 |
| PDM-Open (Privileged) | 2.375 | 0.715 | 2.06 | 5.296 | 0.042 | 85.8 |
| STR(CKS)-300k (Ours) | 2.069 | 1.200 | 2.426 | 5.135 | 0.067 | 82.2 |
| STR(CKS)-16m (Ours) | 1.923 | 1.052 | 2.274 | 4.835 | 0.058 | 84.5 |
| STR(CKS)-124m (Ours) | **1.777** | **0.951** | **2.105** | 4.515 | 0.053 | **88.0** |
| STR(CKS)-1.5b (Ours) | 1.783 | 0.971 | 2.140 | **4.460** | **0.047** | 86.6 |

Table 2: Performance comparison of motion prediction on the validation set of the WOMD.

| Methods | mAP ↑ | minADE ↓ | minFDE ↓ | MR ↓ |
|---|---|---|---|---|
| SceneTransformer (Ngiam et al., 2021) | 0.28 | 0.61 | 1.22 | 0.16 |
| MTR-e2e (Shi et al., 2022) | 0.32 | **0.52** | **1.10** | **0.12** |
| STR(CPS)-16m (Ours) | **0.33** | 0.72 | 1.44 | 0.19 |

the Future States. For better training efficiency, the diffusion decoders are trained separately from the other parts. The details of this decoder and settings for training can be found in Appendix A.1.

## 5.3 SCALING LAWS

We test the scalability of our approach with the NuPlan dataset. We measure the performance with the evaluation loss, which is the sum of the MSE losses for the Key Points and all target future states. The experiment results indicate that the performance of our method smoothly scales over the size of the dataset across 3 magnitudes and over the size of the model across 4 magnitudes. To measure the relation between the size of the dataset, the model size, and the performance, we compare the converged loss value with sufficient training iterations before overfitting for each set of experiments. In general, we find that learning processes on motion prediction and motion planning with our method follow scaling laws similar to the learning processes on language modeling problems. The ratio of each scaling law varies depending on the experimental settings, such as whether or not key point predictions are included. However, the general pattern holds as long as the settings within this group of experiments are the same. The size of the model is measured by the number of trainable parameters of the backbone Transformers.

**Performance with Dataset Size, Model Size, and Computation.** Following (Dauner et al., 2023), we measure the converged loss on the validation 14 NuPlan dataset over the different sizes of the training dataset. We discover a log-log relationship between the log of the evaluation loss and the size of the dataset, as shown in Fig. 2. The size is measured by the number of samples available for the training. This means that as the size of the dataset grows exponentially, the MSE loss will decrease exponentially. This relation promises our method a great scalability to exploit larger datasets. Additionally, we find that larger models generally exploit more from larger datasets.

**Performance with Training Time.** We log the evaluation loss with 4 different model sizes during training. As shown in Fig. 2, larger trajectory models learn significantly faster than the smaller ones before they converge. These curves are the evaluation losses for every 4000 steps while training with ten percent scenarios of the whole training dataset. All 4 experiments in this group are training with the same setting of batch size, learning rates scheduler, and dataset random seed.

## 5.4 QUANTITATIVE ANALYSIS

**Comparison on Motion Planning.** As shown in Table 1, STR [1] surpasses other methods by a large margin. In this table, we compare the performance with diverse approaches. Intelligent Diver

---

[1]This performance of the large model (1.5b) was trained for 0.28 epoch and not fully converged.

Table 3: Performance comparison for different design choices with a subset of the training set. The methods with *CKS* generate future states with 5 auto-regressive Key Points as conditions. The methods with *fwd* and *bkwd* generate these Key Points in forward indices (0.5s, 1s, 2s, 4s, 8s) and backward indices (8s, 4s, 2s, 1s, 0.5s) respectively. Methods indicated by *w/o Diff.* output these Key Points with MLP layers and methods indicated by *w Diff.* output Key Points with diffusion decoders.

| Methods | 8sADE ↓ | 3sFDE ↓ | 5sFDE ↓ | 8sFDE ↓ |
|---|---|---|---|---|
| STR(CS)-16m | 2.223 | 1.253 | 2.608 | 5.480 |
| STR(CKS)-16m fwd w/o Diff. | 3.349 | 2.030 | 4.160 | 7.751 |
| STR(CKS)-16m bkwd w/o Diff. | 2.148 | 1.159 | 2.563 | 5.426 |
| STR(CKS)-16m bkwd w Diff. | **2.095** | **1.129** | **2.519** | **5.300** |

Model (IDM) (Treiber et al., 2000) is a pure rule-based planner that infers future states based on lanes from a graph search algorithm. Similar to our approach, PlannCNN (Renz et al., 2022) also learns from raster representations. Urban Driver (Scheel et al., 2022) and PDM-Open (Dauner et al., 2023) leverage similar model structures and learn from polygon and graph representations. PDM-Open (Privileged) explores additional priors from the centerlines, which are the guidance produced by the IDM planner. Considering learning with additional priors, PDM-Open (Privileged) is not a perfectly fair comparison with other methods. We list its performance with gray font for reference. Compared to these baselines, STR not only outperforms the PlanCNN with similar inputs of raster representations but also exceeds both Urban Driver and PDM-Open by a large margin on both averaged displacement error and final displacement errors. The superior performance indicates STR as a better approach regardless of different representations, or encoders. The larger versions of STR even outperform the PDM-Open (Privileged), saving future methods from picking priors for better performance. For experiments on motion planning with the NuPlan dataset, we implement a simplified version (CKS) that includes only three parts: the context, Key Points, and target states to generate.

**Comparison on Motion Prediction.**   We compare STR with two benchmarks, SceneTransformer (Ngiam et al., 2021) and MTR (Shi et al., 2022) on the validation set of the WOMD. SceneTransformer is a general method for both motion prediction and motion planning, sharing similar model designs with STR by arranging the past states of each agent at each time step in a 2D sequence. On the other hand, STR includes the intention points from MTR in the proposals for classification. We choose the end-to-end version of MTR (MTR-e2e) to facilitate the evaluation of different models under similar input and output conditions, ensuring a fair comparison. As shown in Table 2, STR outperforms both MTR-e2e and SceneTransformer on mAP and comparable results on other metrics without queries and complex refinements at all for the decoding process. These results also reveal the versatile aspects of STR for integrating classification and regression for motion prediction. We implement a simplified version (CPS) that includes only three parts of the whole sequence without Key Points for motion prediction experiments on WOMD.

## 5.5 ABLATION STUDY

We execute the ablation study on the planning task with the NuPlan dataset. We use the STR with 16 million parameters. As shown in Table 3, *STR (CKS) 16m bkwd* outperforms *STR (CS) 16m* showing that adding backwardly generated Key Points enhances the performance in accuracy. However, adding forwardly generated Key Points poisons the performance due to significant accumulative errors. Finally, replacing the Key Points decoder from simple MLP layers with diffusion decoders further improves the performance as shown in the last row.

## 5.6 QUALITATIVE ANALYSIS

Beyond great scalability and better numerical results on accuracy, we explore diverse scenarios with visualizations to further analyze the generalizability of different size STR models. The NuPlan dataset includes scenarios from vast and diverse areas across 4 different cities: Boston, Pittsburgh, Las Vegas, and Singapore. The map topologies across each city exhibit significant variations. For example, there are narrow three-way intersections on the map of Pittsburgh and large parking areas of the hotels on the map of Las Vegas which are absent on the map of Boston.

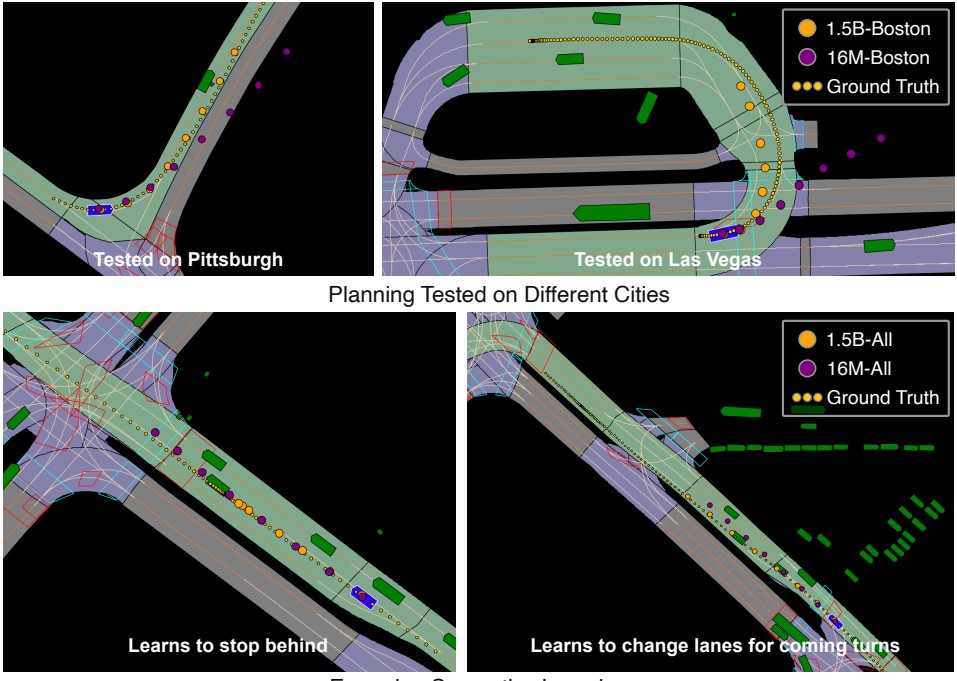

Figure 3: Qualitative analysis on trajectory models of different scales. The route given for each scenario is marked as green roadblocks. The ego vehicle to plan is marked as the dark blue box. All the other road users are marked as green boxes with their given size of shape as well as the yaw angles. The planning results are marked as larger circles in orange for larger models and purple for smaller models. These circles are sampled at every second from the trajectory of 8 seconds in total.

The ability to generalize over a new area or new city not included in the training dataset is crucial for autonomous driving since collecting data is extremely expensive. To examine, we train two models with the sizes of 16 million and 1.5 billion parameters on the NuPlan training dataset collected from Boston. We then evaluate them with the validation dataset in unseen environments, specifically with unique scenarios from Pittsburgh and Las Vegas. Consequently, our findings reveal that larger STR exhibits a significantly better generalization ability when facing a new environment. Additionally, we observe that smaller STR tends to generate an over-smoothed trajectory when traversing an intersection that is different from other intersections in the training dataset, as shown in the example of the upper left case in Fig. 3. The larger 1.5B model also demonstrates enhanced comprehension of unseen road topologies, such as when driving into a hotel pick-up area in Las Vegas which is not present in the training dataset of Boston, as shown in the example of the upper right case in Fig. 3. These examples indicate a better generalization ability of larger trajectory models.

Additional comparison indicates that larger models tend to generate future states with better reasonings about the surroundings, despite similar displacement errors with the smaller ones in these cases. As shown in the lower left case in Fig. 3, the large model learns to stop behind other vehicles. As for the example in the lower right case in Fig. 3, the large model learns to change lanes to the left to prepare for the next coming left turn. Note there are no extra losses or rewards to guide the ego to avoid collisions or commit lane changes in advance.

## 6 CONCLUSION

In this paper, we present a novel way of arranging all elements of both motion prediction and motion planning problems into one sequence and present a general framework, STR. Experimental results indicate that by proper formulation, large trajectory models like STR present great scalability over both the size of the dataset and the size of the Transformers. STR outperforms previous benchmarks on both motion prediction and motion planning on the NuPlan dataset and the WOMD.

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

# A    IMPLEMENTATION DETAILS

## A.1    MODEL AND HYPERPARAMETERS

We use the same model and training settings for both the motion planning on NuPlan and the motion prediction on WOMD. Our transformer backbone is derived from the GPT-2 model from the Huggingface transformers (Wolf et al., 2020). We select SiLU (Elfwing et al., 2018) as the activation function. We use the AdamW (Loshchilov & Hutter, 2018) optimizer with a linear learning rate scheduler with 50 warmup steps, a weight decay of 0.01, and a learning rate of $5 \times 10^{-5}$. We train all models with the same batch size of 16 to compare the performance over different sizes. We elaborate on the details of the model settings for 4 different model sizes we use in our experiments in Table 4. For the motion planning task on the NuPlan dataset, we train the 300k and 16M STR (CKS) on 8 NVIDIA RTX3090 (24G) GPUs. We train the 124M and 1.5B STR (CKS) on 8 NVIDIA A100 (80G) GPUs. For WOMD, we train the model in a 2-stage fashion. We first train the STR (CPS) with the loss of cross-entropy for Proposal classification and MSE for Proposal regression. After converging on the validation set, we train the STR (CPS) with the loss of MSE for the Future States regression. We train the STR (CPS)-16M on 8 NVIDIA RTX A6000 (48G) GPUs for each type of agent.

Table 4: Model settings of the Transformer backbone in different sizes.

| Model Size | Layer | Embedding dimension | Inner dimension | Head |
|---|---|---|---|---|
| 300K | 1 | 64 | 256 | 1 |
| 16M | 4 | 256 | 1024 | 8 |
| 124M | 12 | 768 | 3072 | 12 |
| 1.5B | 48 | 1600 | 6400 | 25 |

## A.2    NUPLAN DATASET CONSTRUCTION

The NuPlan dataset includes diverse real-world driving data across four different cities, which are Boston, Pittsburgh, Vegas, and Singapore. It also provides diverse tags describing each scenario. Each scenario, or each sample in the dataset for training, covers the trajectory and environmental context spans from two seconds in the past to eight seconds in the future. Following the NuPlan challenge's setting, we filter the dataset by 14 different tags. See Fig. 4 for all of these tags and their distributions in the training set. We summarize the number of scenarios of each city after the filtering in Table 5.

## A.3    NUPLAN METRIC AND EVALUATION

NuPlan introduces a unique metric called Open Loop Score (OLS). In this section, we explain the computation details of this metric. There are 5 sub-metrics to compute before the OLS which are average displacement error (ADE), final displacement error (FDE), average heading error (AHE), final heading error (FHE), and miss rate (MR).

The 4 displacement error sub-metrics share the same compute method. Let the 4 displacement errors be $X$={ADE, FDE, AHE, FHE}. The result of each error $x$ is computed by averaging errors across three prediction horizons and can be described as:

$$\bar{x} = \frac{x_3 + x_5 + x_8}{3} \tag{3}$$

Table 5: Number of scenarios of each city in the training set after filtering.

| Boston | Pittsburgh | Singapore | Vegas | Total |
|---|---|---|---|---|
| 941K | 914K | 540K | 12.7M | 15.1M |

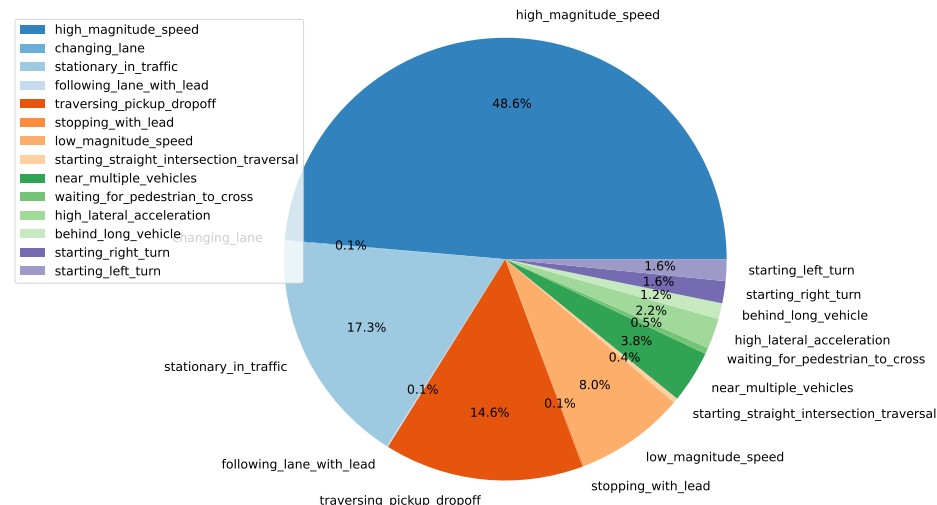

Figure 4: The distribution of the scenarios with each scenario tag in the training set.

$x_3$, $x_5$, $x_8$ are the values of the corresponding errors in horizon-3s (0-3s), horizon-5s (0-5s), and horizon-8s (0-8s) predictions. The score for each sub-metric ranging from 0 to 1 is computed by:

$$score_x = max(0, 1 - \frac{\hat{x}}{threshold_x}) \qquad (4)$$

Specifically, the displacement error threshold is set to 8 (meter) and the heading error threshold is set to 0.8 (rad). The miss rate is the proportion of scenes that are missed as a percentage of all scenes. The score of miss rate $score_{miss}$ is 0 if the scenario-wise miss rate is more than 0.3 else 1.

Finally, the overall weighted score is calculated as the open loop score (OLS) for each scenario:

$$OLS = \frac{w_x * score_x}{\sum_{x \in X} w_x} * score_{miss} \qquad (5)$$

The weights $w_x$ for heading errors are 2 while the weights for displacement errors are 1.

### A.4 NUPLAN RASTER ENCODER

We encoder map elements, routes, traffic lights, and the past states of all road users into different channels of the raster. All the element points are normalized relative positions in the ego-vehicle coordinate. Particularly, the input raster at each timestamp includes 33 channels. The first two channels (0-2) are respectively the route blocks and route lanes. The route blocks are a list of roadblocks, as polygons, provided by the NuPlan dataset for each scenario. The route lanes are all lane center lines that belong to each of these route blocks. The following 20 channels (2-22) are road map channels and each channel encodes one type of road element such as stop lines, roadblocks, various road connectors, and other road components. In the next three channels (22-25), intersection blocks with traffic lights will be rasterized into corresponding channels based on their traffic states which can be green, red, or yellow. In addition to road-related information, at each timestamp in the past two-second horizons, the relative positions and the shapes of all agents are rasterized into the last eight channels (25-33), each channel for each type of agent.

We rasterize all the contexts with two different scopes in the bird's-eye view, as illustrated in Fig. 5. To learn long-term reasoning, we construct low-resolution rasters with an ego agent view field of about 300 meters. To learn low-speed subtle maneuvers, we construct high-resolution rasters with an ego agent view field of about 60 meters. The features of both high-resolution and low-resolution rasters after a ResNet (He et al., 2016) network are concatenated as one observation feature at each time step.

'

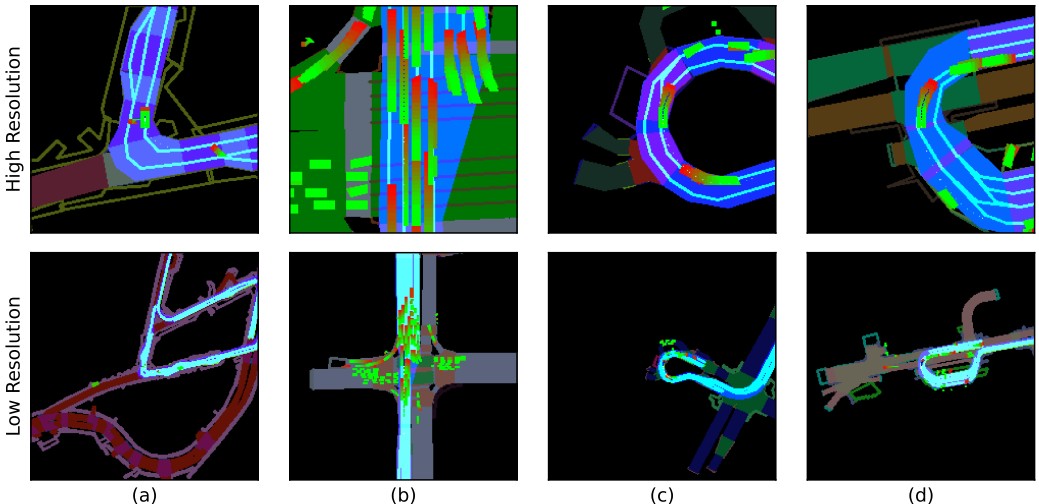

Figure 5: Visualization of rasters in bird view. The upper row is the rasters with high resolution while the second row is the same scenario in low resolution visualization. (a) Start turning left. (b) Traversing intersection (c) Passing roundabout (d) Traversing pickup drop-off

## A.5 DIFFUSION DECODER

For better training efficiency, we train the diffusion-based Key Points decoder separately on the NuPlan dataset. We first train STR with an MLP Key Point decoder with an MSE loss. After converging on the validation set, we freeze the backbone and train the diffusion-based Key Points decoder together with the trajectory decoder. Note the diffusion Key Points decoder still outputs the Key Points one by one in an autoregression style. We expand the embedding dimension and the inner dimension along with the settings of the backbones, as listed in Table 6. For all diffusion Key Points decoders, we use the standard DDPM process (Ho et al., 2020) with 10 steps and a linear noise schedule from $\beta_0 = 0.01$ to $\beta_9 = 0.9$.

Table 6: Model settings of the diffusion Key Points decoder in different sizes.

| Backbone | KP Decoder Size | Layer | Embedding dimension | Inner dimension | Head |
|----------|-----------------|-------|---------------------|-----------------|------|
| 300K | 219K | 1 | 64 | 256 | 1 |
| 16M | 13M | 7 | 256 | 1024 | 8 |
| 124M | 75M | 7 | 768 | 3072 | 8 |
| 1.5B | 0.3B | 7 | 1600 | 6400 | 8 |

## A.6 TRAJECTORY AUGMENTATION

For the task of motion planning on the NuPlan dataset, trajectory augmentations are widely used by most of the previous methods. Following the common augmentation methods provided by the NuPlan devkit, we select a linear perturbation as the only augmentation during training. We add a linearly decreasing noise. This noise is randomly sampled from 0 to 10 percent of the normalized positions 2 seconds ago:

$$\sigma_t = \sigma * \frac{t}{T_m} \tag{6}$$

$T_m$ is the history frames number, $t \in \{T_m, \dots, 0\}$ is the augmenting history frame, where $t = 0$ indicates the current frame. $\sigma$ is the sampled maximum noise.

## A.7 WOMD MOTION PREDICTION

In MTR(Shi et al., 2022), there are mainly three complicated tricks for performance improvement: iterative refinement, local movement refinement and dense future prediction. We only leverage dense future prediction as an auxiliary loss to improve our context encoder but we do not employ the object features updated from it. For multi-modal future trajectory prediction in WOMD, in the inference phase, we explicitly do top $K = 6$ classification on the proposals. After selecting the proposals by our scoring decoder model, we embed the coordinate values respectively and apply the backbone followed by the trajectory decoder to generate a trajectory according to each of the 6 modalities. Our encoder and decoder are much simpler than MTR while keeping comparable performance with its end-to-end results.

## B ADDITIONAL EXPERIMENT RESULTS

### B.1 PER-TYPE RESULTS OF WOMD VALIDATION

The per-category performance of our approach for motion prediction challenges of Waymo Open Motion Dataset is reported in Table 7 for reference.

Table 7: Per-type performance of motion prediction on the validation set of WOMD.

| Object type | mAP ↑ | minADE ↓ | minFDE ↓ | MR ↓ |
|---|---|---|---|---|
| Vehicle | 0.3600 | 0.8964 | 1.7698 | 0.1995 |
| Pedestrian | 0.3274 | 0.3778 | 0.7855 | 0.1239 |
| Cyclist | 0.2880 | 0.8723 | 1.7626 | 0.2447 |
| **Avg** | 0.3251 | 0.7155 | 1.4393 | 0.1894 |

### B.2 QUALITATIVE RESULTS IN WOMD

We provide more qualitative results of our framework in Fig. 6. As shown in the visualization results, our STR model can successfully resolve the problem of uncertainties and multi-modalities. In Fig. 6a, the best proposal is chosen as the indicator of the trajectory generation in the intersection scenario, providing the information of "go straight" instruction and the velocity. As shown in Fig. 6b, our model is able to deal with low-speed scenarios since the velocity states are offered to the backbone. For more complex road situations, our model displays excellent robustness and accuracy in Fig. 6c. Fig. 6d shows the rare "U-turn" case, which certificates the ability of our model to learn from road graphs and inherent patterns in trajectory data.

### B.3 FAILURE CASES IN WOMD

Some failure cases are visualized in Fig. 7. For Fig. 7a and 7b, there are some cases in which the model fails to classify the correct proposals. These are mainly caused by the absence of dynamic traffic information such as traffic lights. To fairly compare with the state-of-the-art models, the dynamic traffic information is not considered in the current STR model, so the model cannot predict a "Stop" or "Slow down" modality properly in the scenarios mentioned above. However, due to the extensive ability of the unified sequence design, the dynamic road graph can be plugged into our model with trivial efforts. The other cases, shown in Fig. 7c and 7d, indicate that although STR successfully classifies the matching proposals, there are some errors generating the future trajectory predictions. These failures are due to the lane-changing behaviors of human drivers, especially when turning in the intersections.

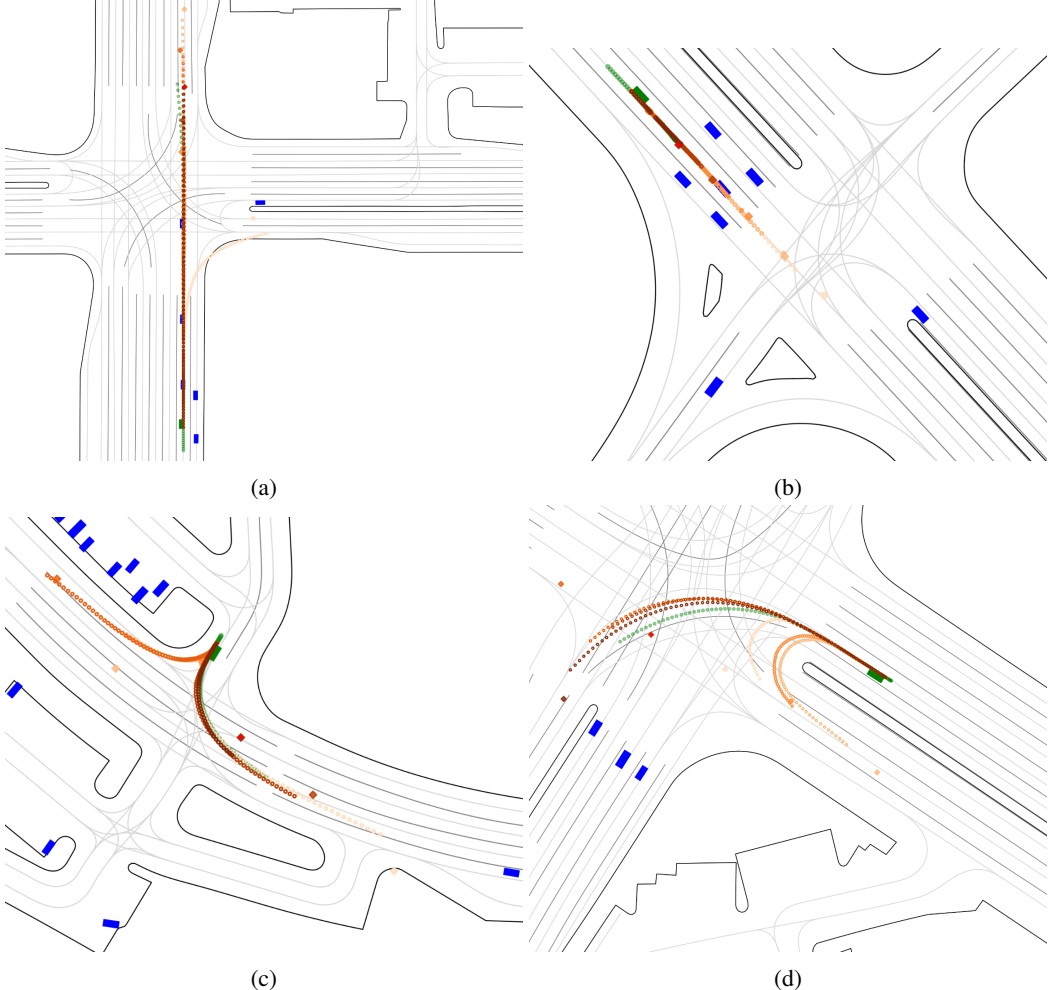

(a)

(b)

(c)

(d)

Figure 6: The qualitative results of motion prediction task in WOMD. The blue rectangles indicate the other traffic agents and the green one is the vehicle to predict. The green dots compose the ground truth trajectory containing both the history frames and the future. The rhombus marks represent the proposals and we use the intensity of colors to indicate the degree of confidence predicted by STR. Correspondingly, each predicted trajectory for each modality is shown in a sequence of dots with the same color as its proposal. The red cross symbol is the ground truth static proposal.

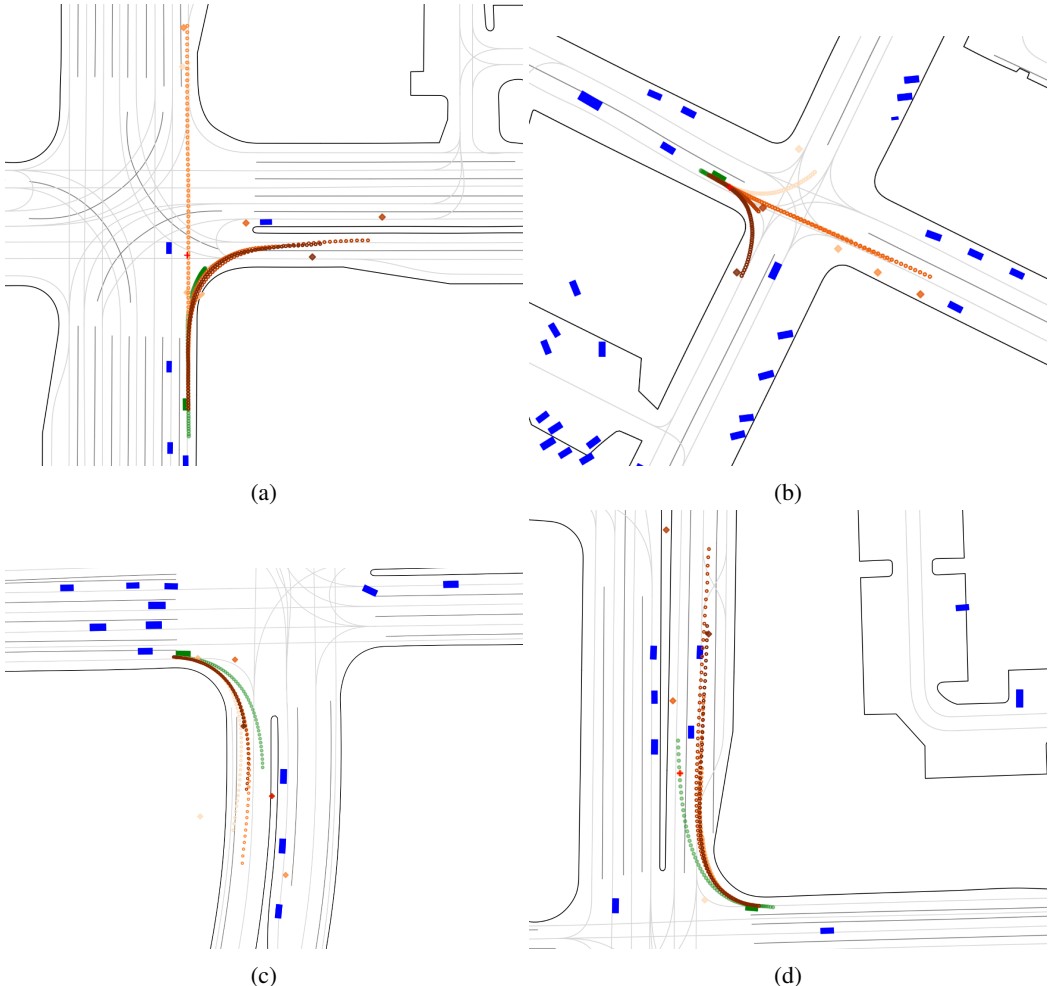

(a)

(b)

(c)

(d)

Figure 7: The visualization of some failure cases in motion prediction task in WOMD. The legend and annotations follow Fig. 6.

