# OpenReview forum: "Large Trajectory Models are Scalable Motion Predictors and Planners"
_ICLR.cc/2024/Conference — Submitted to ICLR 2024_

### Official Review · Reviewer_ixB2 · 2023-10-23

**Soundness:** 2 fair
**Presentation:** 3 good
**Contribution:** 2 fair
**Rating:** 5
**Confidence:** 3

**Summary:**

The paper presents an architecture for trajectory prediction using a generic causal transformer, and validates the scaling law on large datasets (WODM, NuPlan), achieving good performance.

**Strengths:**

1. Sufficient originality and significance. The story captures the existing popular trends, targeting the task of autonomous driving trajectory prediction. It adopts a generic and easily scalable architecture, and validates the scaling law on large datasets, offering a certain reference value.
2. The article is overall well-written, flowing smoothly, and easy to understand.
3. Achieved commendable results on large public datasets.

**Weaknesses:**

1. The story is well told, but the experimental support is weak, some of key ablations are missing:
* There is a lack of ablation studies for the model design, such as quantifying the improvement brought about by the proposed keypoints module, or comparing the GMM decoder versus the diffusion decoder.
* For the planning module, there is a lack of closed-loop evaluation, which is crucial for the planning component.

**Questions:**

1. Why haven't the closed-loop results on NuPlan been included? Since you've already integrated the open-loop evaluation with NuPlan, conducting a closed-loop test should be quite straightforward. Is it because the results were not satisfactory?
2. In Table 2, STR(CPS)-16m (Ours) performs better than MTR-e2e in terms of the MAP metric; however, it lags behind in minADE, minFDE, and MR. Could you elaborate on the reasons behind these differences?

---

> ### Author Response · Authors · 2023-11-14
> **Ablation Study Added**
>
> Thank you for the time to read and review our paper. We find your review very constructive!
>
> We have rewritten the abstract, introduction, and some other parts of the paper to focus on scalability for better clarity. We also attached our full code (with readme and instructions), which will be open-sourced with pre-trained checkpoints for easy reproduction. If you have any further concerns, we would be keen to address them. We will incorporate all of these changes in the final revision.
>
> # General comments
> 1. **Re Ablations on the decoders:** We added a new section 5.5, and Table 3 for the ablation study on the Key Points and the diffusion decoders.
> 2. **Re NuPlan CLS:** We measured the STR(CKS)-16m without diffusion decoders on two CLS metrics. The scores are higher than PDM-Open and lower than PDM-Hybird. We are exploring different flavors of post-procession on the Key Points for better and more stable performance across different test sets. We will include these scores (both CLS-Interactive and CLS-NonInteractive) in the Appendix in the final version.
> 3. **Re Not significantly better than MTR-e2e on all metrics:** As you might notice, we used the STR(CPS)-16m version to compare. We are working on larger models and will include a better performance (on minADE and minFDE) in the final version.

---

> ### Comment · Reviewer_ixB2 · 2023-11-19
> **Mistake in Table 2**
>
> In Table 2, you have incorrectly listed the metrics for SceneTransformer[1]. In the original text, SceneTransformer reports minADE and minFDE using a traditional calculation method that only considers minADE and minFDE at t=8s. However, the metrics you report are calculated by averaging t=3, t=5, and t=8s, which makes the comparison quite unfair. Please make sure to correct this. Refer to MTR[2] (NeurIPS 2022 version) Table 1 for the correction. From this perspective, the results of this paper do not seem solid (with minADE and minFDE being significantly lower than other models) and there is a suspicion of inflated reporting. Additionally, the author has not directly addressed my concerns. After considering the advice of other reviewers, I have decided to temporarily downgrade my rating to ‘weak reject’.
>
> [1] Ngiam J, Caine B, Vasudevan V, et al. Scene transformer: A unified architecture for predicting multiple agent trajectories[J]. arXiv preprint arXiv:2106.08417, 2021.
>
> [2] Shi S, Jiang L, Dai D, et al. Motion transformer with global intention localization and local movement refinement[J]. Advances in Neural Information Processing Systems, 2022, 35: 6531-6543.

---

> > ### Author Response · Authors · 2023-11-20
> > **Table 2 and Section 5.4 Updated**
> >
> > Thank you for your feedback pointing out this mistake in Table 2. We have updated the SceneTransformer result and revised Section 5.4 with a more prudent description of our performance comparison. Note exploring the huge design choice space and reporting a SOTA performance on WOMD was not our contribution. We share our insights with the community on the topic of **scalability** which was scarcely visited by previous works on motion prediction. We understand the importance of numerical comparison. The scaling experiments on the motion prediction task on the WOMD will be included in the final version. All pre-trained checkpoints and codes will be released so that no results in the paper can be inflated.
> >
> >
> > # About Question 2:
> > As requested, we would like to discuss more about lower minFDE and minADE compared with the MTR-e2e. We still do not know exactly why and further exploring exceeds the scope of this paper. However, we would like to point out the result from Table 3 of the MTR [2] paper. Note almost all ablation results have a worse minFDE and minADE than the result from Tabel 1 without more explanations. The minFDE from the same model can easily fluctuate from 1.47 (with closest settings) to 1.37 (best in Table 3) to 1.04 (in Table 1), and the minADE from 0.70 (with closest settings) to 0.52 (best in Table 3) to 0.67 (in Table 1) with different settings. A higher mAP with lower minADE/minFDE or a lower mAP with higher minADE/minFDE are common mismatches for various models on motion prediction metrics.

---

> ### Comment · Reviewer_ixB2 · 2023-11-21
>
> Thank you for your reply.
> I understand that minADE/minFDE and mAP may not necessarily align, I understand that minADE/minFDE and mAP may not necessarily align. What I mean is that such experimental results are not very impressing, and I believe the biggest issue is still the incompleteness of the experiments. As mentioned in your title, motion and planning are the two core tasks you aim to verify, but the experiments for both of these tasks are somewhat lacking. Firstly, there is a lack of scaling data on the WOD motion dataset, and secondly, there is a lack of closed-loop validation for planning. Additionally, the metrics for motion do not seem very impressive (minADE/minFDE are significantly worse than others). I think such experimental results are not sufficiently convincing. I am inclined to maintain the current score.

---

> > ### Author Response · Authors · 2023-11-22
> > **Added quick scaling data on WOMD & Why misaligned CLS relevant?**
> >
> > We implement quick additional scaling experiments on 20% of the training set on WOMD with similar model sizes as we did for the NuPlan dataset. This additional figure indicates similar strong scalability on the evaluation loss during training.
> >
> > Figure link: http://180.167.251.46:880/ICLR_Submission5908/WOMD_quick_scale.jpg
> >
> > As for CLS on NuPlan, we made clear arguments in **Section 5.1, paragraph Metrics for Planning** that CLS is a notable misalignment metric as discussed in the PDM [1] paper. Could you please help us understand why this misaligned metric is so important for us to prove our model's scalability? We think Closed-Loop Scores are clear "better-to-have" results but not compulsory in the scope of this paper.
> >
> > [1] Daniel Dauner, Marcel Hallgarten, Andreas Geiger, and Kashyap Chitta. Parting with misconceptions about learning-based vehicle motion planning. In Conference on Robot Learning (CoRL), 2023.

---

> ### Comment · Reviewer_ixB2 · 2023-11-22
>
> Thank you for the results you've added. I apologize for not making my intentions clear. What I am more interested in seeing is the results of mAP/minADE/minFDE vs scale vs other methods like what you've done in Table1. For planning, as you pointed out, CLS and OLS are not aligned, so when we discuss planning, the final results should be based on CLS rather than OLS. This is because eventually, CLS is what closely represents the real-world scenarios. Otherwise, there would be no difference between the tasks of planning and motion prediction. All the experimental results only prove that Large Trajectory Models are Scalable Motion Predictors but not prove the Scalable Planners.

---

> > ### Author Response · Authors · 2023-11-23
> >
> > Thank you for the constructive advice and clarification.
> >
> > A full scaling analysis like Table 1 on WOMD with other prediction models is interesting but expensive considering the number of experiments. We humbly question how much the value of this analysis would have because:
> > 1. We demonstrate the scalability of STR on WOMD in the previous eval loss figure.
> > 2. We pointed out that MotionLM (the only previous method that did scalability tests) does not scale like STR.
> > We can at least provide the converged loss and mAP/minADE/minFDE results of some other open-sourced methods on some scales in the final version for comparison.
> >
> > Other potential relevant questions:
> > 1. Does eval loss represent scalability or do we need to evaluate on mAP/minADE/minFDE? As discussed previously with reviewer 2D3R, we hope reviewers understand why we chose evaluation loss to measure the scalability rather than mAP/minADE/minFDE in Figure 1. We want to point the reviewers to revisit the scaling law analysis on LLM [1]. They did all measurements on evaluation and test loss for a reason.
> > 2. Scalablity automatically brings SOTA performance (like better than MTR++?) on all metrics on WOMD? As long as our results indicate, this is not the case. Selections of priors (like intention points of MTR) do matter. Post-processions (like NMS) do matter. Ensembling models does matter. All these tricks affect how well our model scales and the final mAP/minADE/minFDE results. For now, there is still a gap and we are exploring which trick is still necessary after we scale. And that is beyond the scope of this paper.
> >
> > For a fair comparison on CLS, we need to adopt our method into the NuPlan planner API and run expensive closed-loop simulations and those take time. We will make sure to include these results and experiment settings with details in our final version.
> >
> > [1] Jared Kaplan, Sam McCandlish, Tom Henighan, Tom B Brown, Benjamin Chess, Rewon Child, Scott Gray, Alec Radford, Jeffrey Wu, and Dario Amodei. Scaling laws for neural language models. arXiv preprint arXiv:2001.08361, 2020.

---

### Official Review · Reviewer_Cy2V · 2023-10-31

**Soundness:** 3 good
**Presentation:** 2 fair
**Contribution:** 2 fair
**Rating:** 5
**Confidence:** 3

**Summary:**

This paper proposes State Transformer (STR) that leverage the transformer model to conduct the motion prediction and motion planning simultaneously, by predicting both the future key points and states.  Extensive experiments show that STR outperforms the baseline methods on NuPlan motion planning dataset significantly.

**Strengths:**

1. The proposed STR is able to conduct motion planning and prediction simultaneously, thereby being able to capture the complexed semantic information.
2. The proposed STR leverages the temporal information by formulating the motion tasks and sequential prediction problems, which helps to improve the performance.

**Weaknesses:**

1. The method description is not well organized and some details are missing, e.g., the proposal, key points and $O_t$ in Fig. 1. Besides, the introduction of the conditional diffusion (Sec 3.2) has no tight logic connection with the method (Sec 4). The only place that mentioned diffusion model in Sec 4 is "STR is compatible with diverse decoders like Gaussian Mixture Model or diffusion decoders."
2. The novelty of this work. Though it is interesting to apply transformer and diffusion on motion planning/prediction tasks, it remains ambiguous as to the unique contributions or innovations introduced by this work in adapting the diffusion model to this specific domain.
Specifically, it is not new to adopt an encoder-decoder architecture to take the context as input, and predict the future states, which has been studied by MPNet [R1]. STR uses a different but existing neural architecture which seemingly lacks new contributions.

[R1] Qureshi, Ahmed H., et al. "Motion planning networks." 2019 International Conference on Robotics and Automation (ICRA). IEEE, 2019.

**Questions:**

1. What is the data format of the context? Do you need to convert it into a set of vectors?
2. As in Section 5.2, you have different choices of encoders for different datasets, what is the concern here? Is there any guidance on how to select the proper encoder when facing a different dataset?
3. Similar to above questions, Sec 4 mentioned that "STR is compatible with diverse decoders like Gaussian Mixture Model or diffusion decoders." Do you have any thoughts on how to select the proper decoders and what are their pros/cons?

---

> ### Author Response · Authors · 2023-11-14
>
> Thank you for the time to read and review our paper.
>
> We simplified the structure and focused on our core insight, which is sequence modeling brings scalability and solves many previous challenges. The design choices of the encoders and decoders are not important in this work. We have rewritten most of the abstract and the introduction to focus on this insight. Please see our uploaded revision.
>
> **Re novelty compared with MPNet:** As we pointed out in the abstract and introduction, learning from large-scale real-world demonstrations brings non-trivial additional challenges compared to [1] which seems to be a more relevant work than MPNet. Specifically, MPNet learns from a small set of RRT* planning results. Additionally, MPNet does not consider other agents (or dynamic environments) at all. Specifically, MPNet does not provide the environment's history, including the past trajectories of other agents and the traffic lights for the motion prediction and planning tasks.
>
> We also attached our full code (with readme and instructions), which will be open-sourced with pre-trained checkpoints for easy reproduction. You can easily check all implementation details about data preprocessing, encoders, and decoders in our code.
>
> We added another section to the related works about previous diffusion decoders on motion planning and motion prediction as an introduction before Section 3.2 and Section 4 for a better continuity flow on writing.
>
> # References
> [1] Yanchao Sun, Shuang Ma, Ratnesh Madaan, Rogerio Bonatti, Furong Huang, and Ashish Kapoor. SMART: Self-supervised multi-task pretraining with control transformers. In International Conference on Learning Representations, 2023.

---

> > ### Comment · Reviewer_Cy2V · 2023-11-20
> > **Thank you for the response.**
> >
> > Thank the authors for addressing part of my questions. I would like to keep my score.

---

### Official Review · Reviewer_2D3R · 2023-11-01

**Soundness:** 3 good
**Presentation:** 2 fair
**Contribution:** 2 fair
**Rating:** 5
**Confidence:** 3

**Summary:**

Inspired by the success of self-attention for sequence modeling in large language models, this paper attempts to demonstrate scaling laws for predicting future vehicle trajectories with self-attention-based large *trajectory* models.

To do so, the paper casts motion forecasting (trajectory prediction for non-ego actors) and motion planning (trajectory prediction for the ego vehicle) as a sequence modeling problem suitable for autoregressive transformers. The sequence tokens are divided between the context, proposal, key points, and future states. The context consists of input data such as the map elements and past actor trajectories. The proposals are only relevant for the Waymo Open Motion Dataset (WOMD) and are (I think) implemented as intention points from MTR++ [0]. The key points are spatial locations 0.5, 1, 2, 4, and 8 seconds in the future. Finally, the future states are the full set of positions over each 100ms frame of the full 8 second trajectory.

The context is generated with dataset-specific encoders. For instance, for NuPlan, a ResNet18 raster encoder is used; for WOMD, an MTR vector encoder is used. The outputs of these encoders become inputs to the autoregressive transformer.

The proposals, key points, and future states have different decoders. Proposals and future states are trained with simple cross-entropy and mean-squared error losses, while the key points are output using a pre-trained keypoint diffusion model.

Their model (State Transformer or STR) is evaluated using standard metrics on NuPlan and WOMD. On WOMD, a 16-million parameter variant of their model is outperformed by MTR [1]. On NuPlan, the top performing model is their 125M parameter model; scaling the model to 1.5B parameters does not significantly improve the model performance. The STR model outperforms all baselines on NuPlan, although there is significantly less literature evaluating on NuPlan (in contrast to WOMD).

Finally, this paper attempts to demonstrate scaling laws and properties of motion forecasting. Qualitative analysis shows that larger models produce improved trajectories, even though quantitative analyses do not show as strong a trend. Additionally, qualitative analyses show improved generalization. The experiments demonstrate improved performance as measured by converged loss as a function of both dataset size and model size.


[0] https://arxiv.org/pdf/2306.17770.pdf
[1] https://arxiv.org/pdf/2209.13508.pdf

**Strengths:**

This paper is the first attempt to scale up trajectory forecasting to models significantly larger than a few million parameters and is one of the first few demonstrations of using an autoregressive transformer for motion forecasting. Additionally, it demonstrates the usefulness of the NuPlan dataset when performing larger scale research, since the dataset is comprised of a larger set of trajectories.

The experiments do convincingly show improved performance with dataset size and model size, indicating that larger datasets and larger models are likely to improve performance significantly further.

**Weaknesses:**

The two key weaknesses of this paper which need to be addressed are complexity and clarity of writing.

First of all, the described model is too general. Instead of describing what was actually implemented, the authors write about what *could* have been implemented. This is partially because the authors are trying to describe their work as one model on two different datasets; in reality, they have implemented two *different* albeit related models, customized towards the two datasets they work on. I would recommend reworking the description to be as clear as possible about what was implemented rather than describing the possibilities. For example:

"These Proposals can be intention points (Shi et al., 2022), goal points (Gu et al., 2021), or endpoints heat maps (Gilles et al., 2021)."

It is not important what the proposals *could* be, what is important is what was actually implemented in this work.

The same applies to key points and future states: please state clearly what *precisely* these are when introducing the concepts, rather than leaving that detail to later sections.

As an example, "For each Key Point, STR is compatible with diverse decoders like Gaussian Mixture Model (GMM) decoders (Chai et al., 2019), or diffusion decoders to produce K different predictions": it does not matter what STR is compatible with, what is important is what was implemented and evaluated. If GMMs were not used, it is not relevant to include in the description.

Next, the model is quite complex, and as a result, it's very hard to understand where gains and performance are coming from. While the authors attempt to reduce motion forecasting to a self-attention transformer sequence modeling problem, in practice, that's not what is happening here; for example, on WOMD, the model is an MTR encoder, followed by a flattening into a sequence, followed by a diffusion model for the keypoints but not for the future states. The parameters are split between the dataset-specific encoders, the transformer backbone, and the diffusion model.

I would recommend making a few changes:

1. Remove diffusion models from this paper. The diffusion models are tangential to the point the paper is aiming to make around scaling, and they add significant complexity to the work that will make it difficult to reproduce and evaluate.
2. Consider separate the model used for NuPlan and for WOMD and describe those models separately.
3. Describe the split of parameters between the encoders and the transformer.
4. (General) Find ways to simplify the setup, model, and description.

Overall, the paper has a very strong core experimental setup and approach, but it is hindered significantly by the complexity of the model (in contrast to the simplicity of large language models) and the complexity of the writing. If these can be simplified to isolate the core result to one about performance of a simple model scaling with data and compute, this paper can be much improved.

**Questions:**

There are a few unclear technical choices made. Causal transformers are relevant for language modeling due to a sampling step after each token. However, in this problem, there is no sampling step until the key point generation, so why is the entire transformer causal? Similarly, autoregressive modeling is crucial for language models due to the sampling step, but since there is no sampling when producing the future states, is it important to have the autoregressive setup?

---

> ### Author Response · Authors · 2023-11-14
>
> Thank you for the time to read and review our paper.
>
> We simplified the structure and focused on our core insight, which is sequence modeling brings scalability, solving many previous challenges. We have rewritten most of the abstract and the introduction to focus on this insight. Please see our uploaded revision.
>
> We also attached our full code (with readme and instructions), which will be open-sourced with pre-trained checkpoints for easy reproducing. Since the scalability comes from the formulation (Note the results in Fig. 2 do not involve the diffusion decoders), readers can easily train a fairly good model without the diffusion decoder’s two-stage training (See new ablation section 5.5).
>
> We tried our best to organize our code for easy training and evaluations with tremendous effort. We used the Hugging Face interface so that importing the dataset or the model is one-line code away. You can scale the model simply by passing different arguments.
>
> If you have any further concerns, we would be keen to address them. We will incorporate all of these changes in the final revision.
>
> # General Comments
> Several misunderstandings in the summary part which led to further concerns later:
> 1. **Why 1.5B STR does not significantly perform better than 125M numerically?** Training 1.5B STR takes more time and the 1.5B STR’s result in Table I is not fully converged with significantly fewer training steps than the 125M STR. We added a footnote (see page 7, marked in red) to further clarify and will include a converged result of the 1.5B STR on NuPlan in the final version.
> 2. **Why do we choose different encoders and decoders for each Dataset?** Scalability is the core of this paper not the design choice of the encoders or decoders for each dataset. To clarify this insight, we rewrote the abstract and the introduction. We observe scalability on both NuPlan and WOMD with both raster and vector encoders. Note adapting additional encoders to a large-scale dataset is a non-trivial task. We will add a vectorized encoder (adapted from the PDM) for NuPlan and a rasterized encoder for WOMD along with their experiment results in the final version.
> 3. **Pre-trained Key Points diffusion decoder?** We did not pre-train the Key Points diffusion decoder. They are trained from scratch in a two-stage fashion, like many text-to-image diffusion decoders.
> 4. **“There is significantly less literature evaluating on NuPlan (in contrast to WOMD)”?** We did spend more literature evaluating experiment results on NuPlan in our paper. For example, the whole 5.3 section about scaling laws is about results on NuPlan. In the 5.4 section, we spent two paragraphs on NuPlan vs. one paragraph on WOMD. The whole 5.5 section, again, is all about results on NuPlan.
>
> Addressing the weaknesses:
> 1. **Re The described model is too general:** We understand your concern. As previously addressed, the study of scalability is the core of this paper. We intentionally test different encoders on different datasets to test if the scalability holds. For better clarity, we deleted other potentials (goal points or heat map) for the Proposal from Section 4 - paragraph Proposal. We also deleted the GMM decoder description for the Key Points from Section 4 - paragraph Key Points. Changes are marked in red in the revised version.
> 2. **Re Not a transformer sequence modeling problem in implementation?** Although we used the MTR encoder, they are not identical. Specifically, we encode the context at each frame separately as illustrated in Fig. 1, but the original MTR encodes the context at all frames together before the decoder. This is a non-trivial change as discussed in [1]. There are no tricks here in the code. We refer the reviewer to check the ‘model.py’ file for more details from the attachment.
> 3. **Re split of parameters:** As introduced in section 5.3, the number of trainable parameters is only from the backbone transformers. Hence, the number of parameters of the encoders is less relevant to the study of the scaling. For your reference, the detailed number of the trainable parameters of each part in our model is listed below.
>
> |      | Mini | Small | Medium | Large |
> | ----------- | ----------- | ----------- | ----------- | ----------- |
> | Waymo Vector Encoder      | 738,736 | 6,337,072 | 52,624,944 | 80,278,960 |
> | NuPlan Raster Encoder      | 11,290,464 | 11,340,672 | 11,474,560 | 11,692,128 |
> | Backbone Transformer.      |  3,332,096 | 16,287,488 | 124,439,808 | 1,557,611,200 |
> | KP MLP Decoder.              | 18,436 | 270,340 | 2,383,876 | 10,291,204 |
> | Key Point Diffusion Decoder | 12,793,700 | 13,410,404 | 16,496,740 | 25,984,868 |
> | Trajectory MLP Decoder | 18,436 | 270,340 | 2,383,876 | 10,291,204 |
>
> # References
> [1] Hao, Y., Song, H., Dong, L., Huang, S., Chi, Z., Wang, W., Ma, S. and Wei, F., 2022. Language models are general-purpose interfaces. arXiv preprint arXiv:2206.06336.

---

> ### Author Response · Authors · 2023-11-21
> **Deadline coming. looking forward to your feedback**
>
> Dear Reviewer 2D3R, we kindly request your feedback as the rebuttal deadline is fast approaching. We hope our previous feedback and revised paper address your concerns and increase your confidence. We would like to thank you again for your review which led this paper to a better quality. We look forward to any additional comments or further discussions.

---

> > ### Comment · Reviewer_2D3R · 2023-11-22
> > **Thank you for the response**
> >
> > Thank you for incorporating the changes and formulating a response. I would like to keep my rating as it is.
> >
> > I think the goal of this work is good and it would be a significant contribution but the experimental setup and results could be stronger to support the conclusion fully. Architecture choices are made which are not clearly justified (diffusion models and causal transformer backbone – why not just scale the Wayformer architecture?) and the results do not indicate significant improvements on metrics from scaling.

---

> > > ### Author Response · Authors · 2023-11-22
> > > **Response and clarification**
> > >
> > > Thank you for your feedback. We appreciate that you see the great value of our insight for the community.
> > >
> > > Could you please help to elaborate on which part of the experimental setup or results fall short of supporting our conclusion? If these requested results are 'better to have', does it make more sense if we add them to the Appendix in our final version?
> > >
> > > **Conflicting feedback**: We are also confused as to why you think the results in Table 1 and Figure 2 do not indicate significant improvements from scaling. These concerns were not brought up before. In fact, you pointed out performance improvements were our strength. Quoting previous review: "The experiments do **convincingly show** improved performance with dataset size and model size, indicating that larger datasets and larger models are likely to improve performance significantly further."
> > >
> > > For architecture choices, we did not use more obvious model designs like Wayformer simply because they **DO NOT SCALE** as well as our proposed STR! This is also the reason why scalability was scarcely studied by previous works. As similar concerns previously pointed out by reviewer 6cJv, a newer model called MotionLM [1] with better parameter efficiency (than Wayformer) and similar autoregressive trajectory prediction failed to scale over 10M trainable parameters (see **Appendix D, paragraph Scaling Analysis**). If you are still confused about the differences between Wayformer and STR, or the logic behind these changes, we strongly recommend checking the detailed feedback we provided previously to reviewer 6cJv.
> > >
> > > [1] Ari Seff, Brian Cera, Dian Chen, Mason Ng, Aurick Zhou, Nigamaa Nayakanti, Khaled S Re- faat, Rami Al-Rfou, and Benjamin Sapp. Motionlm: Multi-agent motion forecasting as language modeling. In Proceedings of the IEEE/CVF International Conference on Computer Vision, pp. 8579–8590, 2023.

---

> > > > ### Comment · Reviewer_2D3R · 2023-11-22
> > > > **Additional clarification**
> > > >
> > > > Figure 2 does clearly show the improvement of the model with data and scale – if you trust the metric of the loss function. However, the loss function does not seem correlated with all the metrics – for example, 8sADE and 3sFDE and 5sFDE are better on the 124M parameter model than the 1.5B model. While this *suggests* that scaling the model is helpful (loss clearly goes down!), it doesn't conclusively demonstrate improved generalization or performance on real-world application, since the metrics that traditionally measure motion forecasting (e.g. 8sADE) actually don't improve.
> > > >
> > > > In the PDF I see Appendix A and B but not D, and no section labeled "Scaling Analysis". Does MotionLM fail to scale above 10M parameters *on the same dataset and training setup* as your study? To make the claim that STR scales better than MotionLM, or better than MTR++ or Wayformer, we need a study that trains those architectures under the same configuration and datasets as STR. To make a strong claim that STR scales better, we need scaling curves on the same dataset for at least one other architecture, while right now scaling curves are only available in this study for STR and not a baseline architecture.

---

> > > > > ### Author Response · Authors · 2023-11-22
> > > > > **Response to 2D3R**
> > > > >
> > > > > Thank you for your reply.
> > > > >
> > > > > 1. About the performance, we have already pointed out in previous feedback 'Why 1.5B STR does not significantly perform better than 125M numerically? ' that the 1.5B STR is not converged at the time. As a result, the 8sADE, 3sFDE, and 5sFDE are not better than the 125M version, which was trained for a much longer period. This will be fixed in the final version. If you review the results in Table 1, there is a clear overall better performance which you cannot ignore for the converged larger models from 300k to 125M. Additionally, we want to point the reviewer to revisit the scaling law analysis on LLM [1]. They did all measurements on **evaluation and test loss** for a reason. Real-world application metrics involve complex encoder, decoder, data augmentation, and loss designs which diverge the focus on scalability. That's why we need to 'trust' more on the loss result rather than ADE/FDEs even though we did show better performance on these metrics with the larger models in Table 1.
> > > > >
> > > > > 2. We were talking about Appendix D of the **MotionLM** paper, not our paper. We did not analyze the scalability of MotionLM since their codes are not open-sourced for reproduction. Just in case the reviewer misses the paper again, there is a link to the MotionLM paper below. Please DO READ the Appendix D of it to acknowledge that they did fail to scale above 10M. The scalability is not related to different datasets. Just like there is no need to test the scalability of GPT on every single dataset. If you still worrying about our scalability on the WOMD, here is a quick analysis we did for the reviewer ixB2 in the following figure. We implement quick additional scaling experiments on the 20% of the training set on WOMD with similar model sizes as we did on the NuPlan dataset. This additional figure indicates similar strong scalability on the evaluation loss during training. We could have done the additional experiments you asked for if you brought them in the first review. As the rebuttal is ending, we will find some models (open-sourced) to compare the scalability on the same subset with the same training settings in the final version.
> > > > >
> > > > > Figure link: http://180.167.251.46:880/ICLR_Submission5908/WOMD_quick_scale.jpg
> > > > >
> > > > > MotionLM paper link: https://arxiv.org/abs/2309.16534
> > > > >
> > > > > [1] Jared Kaplan, Sam McCandlish, Tom Henighan, Tom B Brown, Benjamin Chess, Rewon Child, Scott Gray, Alec Radford, Jeffrey Wu, and Dario Amodei. Scaling laws for neural language models. arXiv preprint arXiv:2001.08361, 2020.

---

### Official Review · Reviewer_6cJv · 2023-11-02

**Soundness:** 2 fair
**Presentation:** 2 fair
**Contribution:** 3 good
**Rating:** 5
**Confidence:** 3

**Summary:**

The authors tackle the problems of motion prediction and planning for self-driving. The authors first frame these problems as identical except for the fact that planning requires conditioning on a high-level route. The authors then propose to model 8 second trajectories by autoregressively sampling locations at (8,4,2,1,0.5) seconds into the future, then using regression to upsample these key points to a full 10hz 8 second trajectory. A highlight of the paper is that the authors leverage causal transformers and demonstrate that larger variants of their model converge to a better validation loss as the dataset size is scaled, similar to large language models.

**Strengths:**

Motion prediction is certainly an important problem for self-driving, and I think stating explicitly how planning and motion prediction are connected as the authors have done is important for the self-driving community. I also think understanding how these models scale with compute and data is important so that we can make better guesses about how much data needs to be collected in order to achieve a certain performance. I haven't seen these kinds of scaling plots in other motion prediction papers. Finally, I think diffusion is a promising technique for modeling trajectory distributions, and it's great to see new techniques - such as the author's approach of only modeling a coarse trajectory then regressing the rest - for making diffusion work in practice for the motion prediction task.

**Weaknesses:**

The first weakness I want to mention is that I'm missing the motivation for some of the design decisions of the model. A central claim of the paper is that they reduce motion prediction to a "sequence modeling task". However, for the historical trajectories and map information that the model conditions on, I don't see the motivation for encoding this information causally in time. Full attention across time should work just as well while being more parameter-efficient (for instance, as was done in wayformer), especially when some of the history is partially occluded as in motion prediction data. Additionally, since the authors use diffusion to model future keypoints, I also don't see the motivation to force the diffusion model to be autoregressive. In my mind, the diffusion model should be able to model all future points at once without generating each one-by-one. If performance was amazing with these choices, I'd understand it, but Table 2 shows that motion prediction performance is well below SOTA. If the authors want to claim that these choices improve the scalability of the model, they should compare against a model with full attention in the encoder and all-at-once diffusion in the scaling plot Figure 2.

The second weakness I see is that the way in which the model is adapted from MTR or other motion prediction models is not stated as clearly as it should be. My sense is that the method that the authors propose involves training a GPT-like model with temporally-causal mask to regress (x,y) locations at (8,4,2,1,0.5) seconds, then freezing the encoder and training the keypoint decoder with diffusion instead of regression, then keeping the encoder frozen and training an output head that upsamples the keypoints to a 10hz 8-second trajectory. The authors also reference "classification scores" though, so I don't think I'm getting the full picture. My sense from Appendix A.1 is that the reason for these choices is that they stabilize the training of the diffusion model. If that's the case, I think most of the motivation of the paper should emphasize that the goal is to improve stability of diffusion for motion prediction, not to turn motion prediction to a sequence prediction task which is currently stated as the main motivation.

Other notes are included below:

- abstract "learn to make complex reasonings for long-term planning, extending beyond the horizon of 8 seconds" - where do the authors show that the model generalizes beyond 8 seconds in the paper?
- intro "We arrange the past and future states of the target into one sequence for learning and generation" - I feel the main novelty of the paper is in the design of the decoder. So maybe it's only important to mention that future states are encoded in a sequence.
- 3.1 first paragraph - there's a mismatch in notation between $s^F$ and $s^T$. Additionally, it should probably be acknowledged in this paragraph that this paper approximates the multi-agent future trajectory distribution as independent, e.g. $p(s^T | c, s_{ego}, s) \approx p(s^T_{ego} | c, s_{ego}, s) \cdot p(s^T_0 | c, s_{ego}, s) \cdot ... \cdot p(s^T_N | c, s_{ego}, s)$.
- Section 4 "Finally, the model learns the pattern to generate all future states given the contexts and conditions with direct regression losses". Would it make sense to represent the full trajectory as a sequence of key points? Regression can lead to out-of-distribution trajectories, even when key points at t=(0.5,1,2,4,8) seconds are used for conditioning.
- Figure 2 - for the plot on the right, what do the circular dots and the square/triangle dots represent?
- 5.2 "in case the potential features from images and videos from cameras are able to be incorporated in the future" - I don't exactly see why it's easier to incorporate camera-frame features if the input is rasterized? Some elaboration of this point would be helpful.
- 5.2 "decoders" - I found this section hard to understand. What are "stacked MLPs"? What is the "additional MSE loss" in addition to? Is "distance regression loss" different from MSE loss? For the "classification scores", what is the network classifying between?
- Appendix A.1 "a plain MLP decoder is used when training the Transformer backbone from scratch, then the backbone is frozen and we train a diffusion-based key points decoder" are there issues when the model is trained with diffusion from scratch? Is there a reason to make the decoder autoregressive if diffusion is being used?
- Table 2 - Can the authors expand on why they only compare to MTR-e2e? MTR and MTR++ get mAP above 0.40 on WOMD

**Questions:**

My main questions are included in the "weaknesses" section. If the authors only focused on comparing scaling laws for standard motion prediction architectures, that would be one thing. But since the authors measured scaling curves only for the new model that they proposed, it's important to understand if the proposed model has a principled design. It seems the authors main some design choices to improve the stability of diffusion, and it would be very helpful if the authors could ablate these choices more thoroughly. I also currently do not see the value in framing the approach as "sequence modeling" - which is a big emphasis of the paper - given that full attention and all-at-once diffusion should be more compute and parameter efficient.

**Details Of Ethics Concerns:**

None.

---

> ### Author Response · Authors · 2023-11-14
>
> Dear Reviewer 6cJv:
>
> Thank you for the time to read and review our paper. We find your review very constructive and in great detail!
>
> These questions and concerns lead us to reconsider the structures to present our core insight, which is sequence modeling brings scalability and scaling up solves many previous challenges. We have rewritten most of the abstract and the introduction to focus on this insight. Please see our uploaded revision.
>
> We also attached our full code (with readme and instructions) which will be open-sourced with pre-trained checkpoints for easy reproducing. Since the scalability comes from the formulation (Note the results in Fig. 2 do not involve the diffusion decoders), readers can easily train a reasonably good model without diffusion’s two-stage training.
>
> If you have any further concerns, we would be keen to address them. We will incorporate all of these changes in the final revision and credit this anonymous reviewer in the acknowledgment.
>
> # General Comments
>
> 1. **Re Missing the value of sequence modeling formulation:** We find that sequence modeling formulation is the key to scalability. Our experiments indicate great scalability, whether with or without the diffusion decoders on both NuPlan and WOMD. So, the choices for neither encoders nor decoders are relevant to the scalability. We also added a comparison with the recent MotionLM [1] paper, indicating fundamental scalability differences with previous non-sequence modeling models. (see the revised version, Section 1 - Intro, paragraph 3 marked in red). MotionLM also used autoregressive down-sampled (2Hz) key points for prediction but they cannot scale over 10M.
>
> 2. **Re Full attention without masks?:** We did implement a mask for invalid data of the WOMD dataset. We skip the details because this is a common implementation for the WOMD dataset only. We believe more shapes of attention for better parameter efficiency are not relevant to scalability and we decided to leave further experiments to future works.
>
> 3. **Re Why Autoregressively Diffusion (1by1) for the Key Points?:** We tried diffusion decoders in both 1-by-1 and 5-at-once styles. The results are almost the same across all metrics, as shown in the table below. Since the 1-by-1 generation provides more options for post-processing, we chose this setting as default.
>
> |      | 8sADE  | 3sFDE | 5sFDE | 8sFDE |
> | ----------- | ----------- | ----------- | ----------- | ----------- |
> | STR(CKS) 16m Bwd w Diffusion (1by1) | 2.095 | 1.129 | 2.519 | 5.300 |
> | STR(CKS) 16m Bwd w Diffusion (AllAtOnce) | 2.107 | 1.140 | 2.535 | 5.327 |
>
> 4. **Re Diffusion Training in 3 stages?:** There are only two stages for training the diffusion decoders. It is a common training pipeline like training text-to-image diffusion models. The diffusion decoder is trained together with the trajectory decoder in the second stage. We revise and elaborate on this detail in the new Appendix A5 section.
>
> # Comments on Minors:
>
> 1. **Re Long-term reasonings in the abstract:** This is illustrated with the lower right example in Fig. 3. The vehicle changed to a left lane to prepare a left turn coming next and the turning happened beyond 8 seconds in the future. We will add more examples and discussion to the Appendix in the final version.
> 2. **Re Writings in the Intro:** We rephrased the paragraph to avoid the ambiguities you pointed out.
> 3. **Re 3.1 S^F:** Revised.
> 4. **Re Section 4:** Revised for better clarity. See the paragraph on “​​Future States” in the revised version.
> 5. **Re Figure 2 Legends:** Legends added. Thank you.
> 6. **Re Section 5.2 - Rasterize help to learn camera features?:** Revised. Changes were marked in red in Section 5.2 - Paragraph Encoder.
> 7. **Re Section 5.2 - Writing issues about the Decoder paragraph:** We rewrote the whole paragraph. See the paragraph on “Decoders” in the revised version.
> 8. **Re Training details of the diffusion decoders:** Addressed in General Comments 3.
> 9. **Re Compare with MTR (not e2e) and MTR++:** The MTR (not e2e) adds NMS and other non-learning tricks to boost the performance, making it not a fair comparison. However, we are working on comparing the larger STRs with them on the test set of WOMD (on the leaderboard). The results will be included in the final version.
>
> # References:
>
> [1] Ari Seff, Brian Cera, Dian Chen, Mason Ng, Aurick Zhou, Nigamaa Nayakanti, Khaled S Re- faat, Rami Al-Rfou, and Benjamin Sapp. Motionlm: Multi-agent motion forecasting as language modeling. In Proceedings of the IEEE/CVF International Conference on Computer Vision, pp. 8579–8590, 2023.

---

> ### Author Response · Authors · 2023-11-21
> **Deadline coming. looking forward to your feedback**
>
> Dear Reviewer 6cJv, We kindly request your feedback as the rebuttal deadline is approaching in less than two days. We hope our previous feedback addresses your concern. We would like to thank you again for your time and previous review and we are looking forward to further discussions.

---

> > ### Comment · Reviewer_6cJv · 2023-11-22
> > **Thank you for the response**
> >
> > - "MotionLM also used autoregressive down-sampled (2Hz) key points for prediction but they cannot scale over 10M." - the MotionLM architecture is very parameter efficient, but that doesn't mean it "cannot scale over 10M". With more data, it would become optimal to train a larger model and that would be very doable. In fact, being more parameter efficient indicates better scalability in my mind.
> > - "scalability differences with previous non-sequence modeling models" - isn't MotionLM also a sequence modeling model?
> > - "We did implement a mask for invalid data of the WOMD dataset." - the mask I'm curious about is the causal mask which would make the transformer sequential. Do the authors use a causal mask for the attention in the encoder for historical trajectories?

---

> > > ### Author Response · Authors · 2023-11-22
> > > **Response and clarification**
> > >
> > > Thank you so much for your feedback. We want to clarify your remaining concerns.
> > > 1. **Re MotionLM does or at least potentially scale?**: It is not our words saying that the MotionLM cannot scale over 10M. We point the reviewer to **Appendix D, paragraph Scaling Analysis** of that paper. In this paragraph, the authors clearly pointed out that they did try to scale their model to 27M parameters. The experiment results indicate that the model overfits the training set with worse performance on validation. This conclusion also aligns well with our previous experience when scaling other motion prediction models. Additionally, we believe there are no previous works that discussed the relation between scalability and parameter efficiency under **the context of motion prediction**. We cannot make that assumption since MotionLM can only scale to about 10M with a very parameter-efficient structure. Previous works also found their model hard to scale on the larger NuPlan dataset, since the size of the dataset might not be the issue. We can definitely provide futher scaling experiments with more methods on NuPlan to the Appendix to clarify this in the final version.
> > > 2. **Re MotionLM is also a sequential modeling model?**: We find this reviewer is confused about sequential modeling models, similar to the reviewer 2D3R below. We want to point the reviewer to the MetaLM [1] for better clarification. Specifically, please check **Figure 3 on Page 6**. As you can see, there is a fundamental difference between the "Prefix LM" ( Encoder-Deocder Structure) and Semi-Causal LM structure. To be more specific in the context of motion prediction, it is important to include the past states **separately** in the same sequence for the backbone. There is also a great parallel example just to clear up any more concerns. In the problem of text classification, one does not put all the text of the sample into one token for the backbone transformer. Instead, one puts each token next to the other as a sequence into the backbone, even none of them has direct supervision during the training. The previous method compressed all information into one token (or embedding if you do not need tokenizations) which is an encoder-decoder structure but the latter method is a sequence modeling structure. Please correct us if we misunderstand your concern. And also please indicate if you still feel confused about the difference after our explanation or about the previous text classification example.
> > > 3. **Re causal mask for the attention in the encoder for historical trajectories?** : For WOMD and NuPlan, we implement a single Multilayer Perceptron (MLP) layer followed by an activation layer of Tanh to encoder (the brown encoder for S in Fig. 1) historical trajectories (named states in our paper) for the ego (or target) agent. If you are asking about encoding the historical trajectories of the other agents (the blue encoder for O in Fig. 1), neither the STR nor the MTR encoder uses a causal mask for the attention in the encoder. Additionally, it does not make sense for the STR to use a causal mask in the encoder since at each time step, there is only one state at that time step for each agent. Please correct us if we misinterpret your question.
> > >
> > > [1] Hao, Y., Song, H., Dong, L., Huang, S., Chi, Z., Wang, W., Ma, S. and Wei, F., 2022. Language models are general-purpose interfaces. arXiv preprint arXiv:2206.06336.

---

### Meta-Review · Area_Chair_xsat · 2023-12-06

**Metareview:**

Summary: This paper investigates if scaling laws apply to large trajectory forecasting models. It scales up forecasters to 1.5B parameters and demonstrates the utility on the nuPlan and WOMD large dataset.

Strengths: Investigating the scaling laws of trajectory predictors is very timely, and the paper provides a comparison to the SOTA model MotionLM, which also is focused on scaling up large trajectory predictors.

Weaknesses: Reviewers raise the need for evaluating the model in closed-loop simulation to understand the potential benefits during deployment of the proposed STR, and raise the desire for more clarity in the writing (e.g., description of the proposed approach and disentangling experiments on WOMD and NuPlan) and raise a desire to understand where exactly the performance improvements come from (e.g., is it the model encodings / architecture vs. capacity vs. dataset size). Reviewers also raised questions about why sometimes the proposed approach was better that SOTA and other times it was not (e.g., MTR-e2e performs better than the proposed approach 16M model).

**Justification For Why Not Higher Score:**

Due to the unclear results across the board compared to prior models, I chose not to give a higher score.

**Justification For Why Not Lower Score:**

N/A

---

### Decision · Program_Chairs · 2024-01-16

Reject